# Spatio-temporal modelling of the first Chikungunya epidemic in an intra-urban setting: The role of socioeconomic status, environment and temperature

**Laís Picinini Freitas**[1,2], **Alexandra M. Schmidt**[3], **William Cossich**[4], **Oswaldo Gonçalves Cruz**[2], **Marilia Sá Carvalho**[2]*

**1** Programa de Pós-Graduação em Epidemiologia em Saúde Pública, Escola Nacional de Saúde Pública Sergio Arouca (ENSP), Oswaldo Cruz Foundation, Rio de Janeiro, Brazil, **2** Programa de Computação Científica (PROCC), Oswaldo Cruz Foundation, Rio de Janeiro, Brazil, **3** Department of Epidemiology, Biostatistics and Occupational Health, McGill University, Montreal, Canada, **4** Department of Physics and Astronomy, University of Bologna, Bologna, Italy

\* marilia.carvalho@fiocruz.br

**Data Availability Statement:** All data analysed during this study are secondary data and are publicly available. The data on chikungunya cases

## Abstract

Three key elements are the drivers of *Aedes*-borne disease: mosquito infestation, virus circulating, and susceptible human population. However, information on these aspects is not easily available in low- and middle-income countries. We analysed data on factors that influence one or more of those elements to study the first chikungunya epidemic in Rio de Janeiro city in 2016. Using spatio-temporal models, under the Bayesian framework, we estimated the association of those factors with chikungunya reported cases by neighbourhood and week. To estimate the minimum temperature effect in a non-linear fashion, we used a transfer function considering an instantaneous effect and propagation of a proportion of such effect to future times. The sociodevelopment index and the proportion of green areas (areas with agriculture, swamps and shoals, tree and shrub cover, and woody-grass cover) were included in the model with time-varying coefficients, allowing us to explore how their associations with the number of cases change throughout the epidemic. There were 13627 chikungunya cases in the study period. The sociodevelopment index presented the strongest association, inversely related to the risk of cases. Such association was more pronounced in the first weeks, indicating that socioeconomically vulnerable neighbourhoods were affected first and hardest by the epidemic. The proportion of green areas effect was null for most weeks. The temperature was directly associated with the risk of chikungunya for most neighbourhoods, with different decaying patterns. The temperature effect persisted longer where the epidemic was concentrated. In such locations, interventions should be designed to be continuous and to work in the long term. We observed that the role of the covariates changes over time. Therefore, time-varying coefficients should be widely incorporated when modelling *Aedes*-borne diseases. Our model contributed to the understanding of the spatio-temporal dynamics of an urban *Aedes*-borne disease introduction in a tropical metropolitan city.

can be downloaded at the Rio de Janeiro Municipal Secretariat of Heath website at http://www.rio.rj.gov.br/web/sms/exibeConteudo?id=4769664. Population, sociodevelopment index and land use data from the Instituto Pereira Passos can be found at www.data.rio. Temperature data were obtained from the Brazilian National Institute of Meteorology (https://tempo.inmet.gov.br/), the Brazilian Airspace Control Department (https://www.redemet.aer.mil.br/?i=produtos&p=consulta-de-mensagens-opmet), the Rio de Janeiro State Environmental Institute (http://200.20.53.25/qualiar/home/index), the Rio de Janeiro Municipal Environmental Secretariat (http://www.data.rio/datasets/dados-hor%C3%A1rios-do-monitoramento-da-qualidade-do-ar-monitorar?orderBy=Data) and the Alerta Rio System (http://alertario.rio.rj.gov.br/download/dados-meteorologicos/), and can be downloaded at their websites. However we made the co-variables datasets available in the format used for the statistical analysis at https://github.com/laispfreitas/ICAR_chikungunya.

**Funding:** This study was partially supported by the Coordination for the Improvement of Higher Education Personnel (Coordenação de Aperfeiçoamento de Pessoal de Nível Superior - CAPES) - Finance Code 001, to MSC and LPF. LPF received funds from the Emerging Leaders in the Americas Program (ELAP), Government of Canada. AMS acknowledges the support of the Natural Sciences and Engineering Research Council (NSERC) of Canada (RGPIN-2017-04999). MSC received grants from Fundação Carlos Chagas Filho de Amparo à Pesquisa do Estado do Rio de Janeiro (FAPERJ, grant n° E_26/201.356/2014) and support from Conselho Nacional de Desenvolvimento Científico e Tecnológico (CNPq, grant n° 304101/2017-6). The funders had no role in study design, data collection and analysis, decision to publish, or preparation of the manuscript.

**Competing interests:** The authors have declared that no competing interests exist.

## Author summary

Viruses transmitted by the *Aedes* mosquitoes represent a major public health concern. With the abundance of the mosquito and susceptible human population, the entry of new *Aedes*-transmitted viruses brings the risk of large epidemics. The first-ever chikungunya epidemic in Rio de Janeiro city, Brazil, happened in 2016. We used neighbourhood information on the environment, socioeconomic status, and weekly temperature, to study the disease spread within the city. Our results show that better socioeconomic status plays a major role in preventing the disease, with poorer areas being affected first and harder by the epidemic. This highlights that improving sanitary and socioeconomic conditions is essential for *Aedes*-borne diseases prevention and control. The temperature increased the risk of chikungunya cases, and this effect persisted for longer in areas where the epidemic was concentrated. This indicates that interventions should be designed to be long-lasting in such locations. Our results contribute to understanding better the dynamics of a first urban *Aedes*-borne disease epidemic in a tropical metropolitan city, with the potential to help design better interventions for disease prevention and control.

## Introduction

The first chikungunya virus (CHIKV) epidemic in Rio de Janeiro city, the second most populated city in Brazil and its leading tourist destination, occurred in 2016 [1]. CHIKV is transmitted to humans by the same vectors as dengue viruses (DENV), the *Aedes* mosquitoes [2]. Vector-control activities have not prevented Rio de Janeiro from being endemic for dengue for years, nor from having experienced large dengue epidemics every three to four years, in general [3–5].

For a chikungunya epidemic to occur, three main elements are necessary, represented by the blue area in Fig 1: mosquito population, susceptible human population, and the virus circulating [6–8]. The *Ae. aegypti* mosquito is present all over the city of Rio de Janeiro, facilitating the establishment of a new arbovirus. Because CHIKV and DENV belong to different families, previous immunity to DENV does not protect against CHIKV infection, and the population of Rio de Janeiro can be considered equally naïve to CHIKV before 2016. Therefore, given the presence of the mosquito population and susceptible human population, the occurrence of local CHIKV transmission in Rio de Janeiro was conditioned by the entry of the virus. Once the virus is circulating and established, how it will spread within the city depends on multiple factors. As a consequence, some areas of the city will experience the epidemic at different times, and will also have different attack rates.

Reliable data regarding the intra-urban level of vector population, the susceptible human population over time, and the entry time of the virus in each location are necessary to understand the spatio-temporal dynamics of an epidemic. However, these data are not usually available. Alternatively, measured factors–such as temperature, socioeconomic and environmental factors–indirectly associated with the number of chikungunya cases (represented by red arrows in Fig 1) can be used as proxies. These factors have a direct effect (represented by black arrows in Fig 1) on the necessary elements for an epidemic to occur (inside the blue area), which are unmeasured. The temperature varies with time, and also present different effects on the epidemic over time. Socioeconomic and environmental characteristics take long periods to show important changes and can be considered fixed during the period of an epidemic.

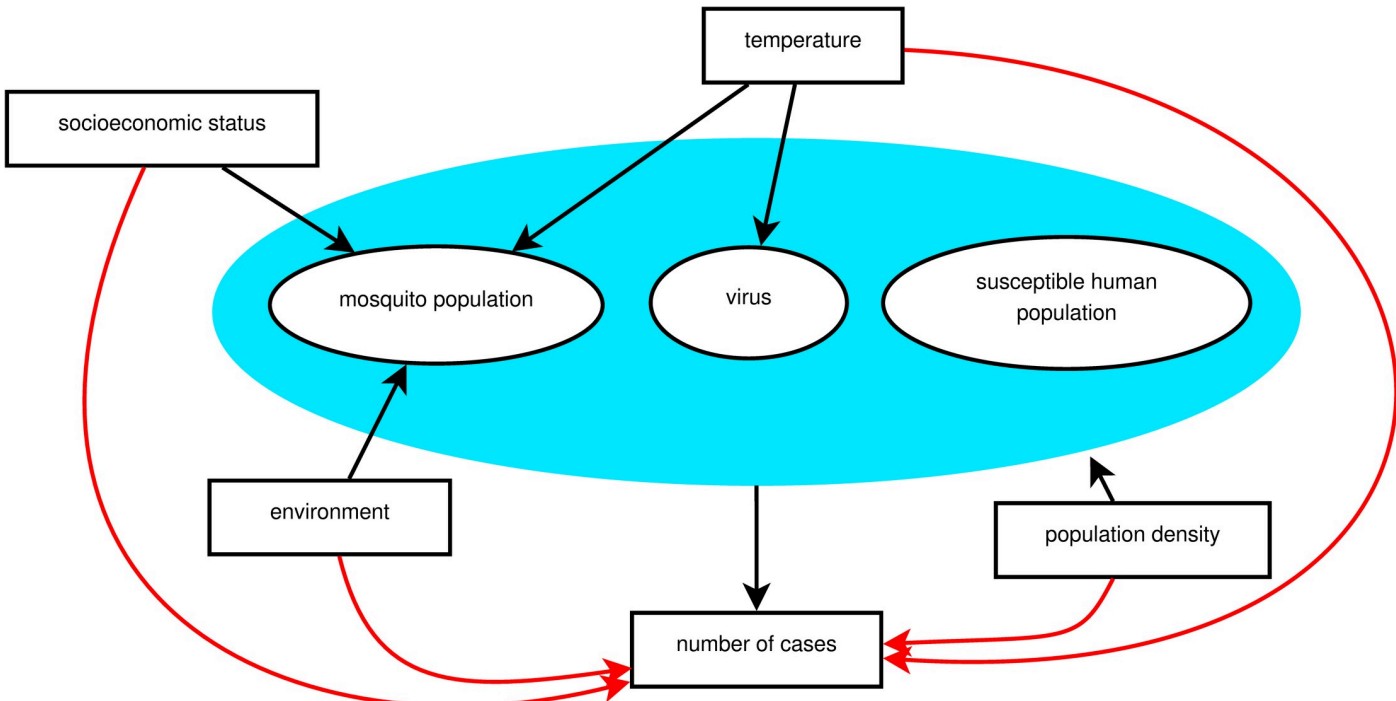

**Fig 1. A theoretical model for a chikungunya epidemic in a given region.** Direct associations are represented by black arrows and indirect associations by red arrows. The blue area includes the necessary elements for the epidemic to occur.

However, the way these factors impact the number of cases can change as the epidemic progresses, i.e., with time-varying effects.

The temperature affects the *Ae. aegypti* population, the virus, and the mosquito-virus interaction [9]. Ideal temperature conditions accelerate all stages of the mosquito life cycle, increasing the population in the long term. In the short term, the temperature influences the mosquito activity as well as the length of the virus incubation period, with maximal transmission occurring around 26–29°C [10]. Regarding the environment, the *Ae. aegypti* mosquitoes are highly adapted to urban settings, and the level of urbanisation is inversely correlated with the proportion of green areas [11]. In Rio de Janeiro city, previous studies found the *Ae. aegypti* mosquito to be more abundant in urban locations compared to rural and forested areas [12,13]. The socioeconomic status impacts the mosquito population as disorderly urbanisation and inadequate sanitary conditions favour the presence of the mosquito most common reproduction site: containers filled with water found inside or in the surroundings of domiciles [12,14]. Furthermore, high population densities favour contact between mosquitoes and humans, increasing the chance of infection and transmission [15].

In the last decades, models that take into account the spatial dependency structure of the cases have been applied to better estimate covariates' association with *Aedes*-borne diseases epidemics [16–19]. The application of spatial models for intra-urban settings is growing more recently [20–23]. These model types, including intrinsic conditional autoregressive (ICAR) models [24], are built under the assumption that adjacent areas share similar characteristics. The inclusion of a latent spatial random effect accounts for both the spatial structure and unmeasurable factors [25]. The inclusion of time-varying coefficients allows us to explore how the effect of the covariates change throughout the epidemic. The temperature is usually included in statistical models for *Aedes*-borne diseases in a linear fashion and with a pre-

defined lag. We propose to estimate the temperature effect in a non-linear framework using a transfer function, including an immediate effect and a memory effect that propagates to future times. Another advantage of using a transfer function is that the estimation of the lag of the effect is data-driven [26].

We used this methodological approach (ICAR models with time-varying coefficients and a transfer function) to identify how temperature, socioeconomic and environmental factors are related to the space-time progression of an *Aedes*-borne disease epidemic in an intra-urban setting. The first chikungunya epidemic in Rio de Janeiro, a large tropical city with environmental and socioeconomic disparities, presents ideal conditions to this end.

## Methods

### Ethics statement

This study was approved by the Research Ethics Committee of Escola Nacional de Saúde Pública Sergio Arouca (ENSP)–Fundação Oswaldo Cruz, approval number 2.879.430. Informed consent was not required as this is a study using secondary data and the data were analysed anonymously.

### Study site

Rio de Janeiro is the second-largest city in Brazil, with 6.3 million inhabitants (2010), and its primary tourist destination. Rio's area is 1204 km$^2$, with 160 neighbourhoods grouped into four large regions (Downtown, South, North and West). These regions are subdivided in 10 health districts called programmatic areas: area 1.0 (Downtown region); areas 2.1 and 2.2 (South region); areas 3.1, 3.2, 3.3 (North region); and areas 4.0, 5.1, 5.2 and 5.3 (West region) (Fig 2).

With three mountain massifs and 84 km of beaches, Rio has a diverse geography that is directly associated with the history of occupation and with socioeconomic disparities [27]. The Downtown region is the historical, commercial and financial centre of the city, with many cultural establishments. The South region is the most popular tourist destination, with famous beaches and wealthy neighbourhoods. In the North region, there are very large slums ("*favelas*") and nearly 27% of the population, almost 2.4 million people, live in such communities [28]. The West region has more heterogeneous characteristics among its neighbourhoods, being the area 5.1 more densely populated, areas 5.2 and 5.3 less urbanised, and area 4.0 wealthier.

### Data

**Chikungunya cases.**    In Brazil, all suspected chikungunya cases attending a health care facility must be notified to the Ministry of Health. This is done by a health care worker–usually, the physician–filling a form at the Notifiable Diseases Information System (*Sistema de Informação de Agravos de Notificação*–SINAN).

We obtained data on chikungunya cases from SINAN via the Rio de Janeiro Municipal Secretariat of Heath and are publicly available [29]. The data corresponded to notified cases of chikungunya (confirmed by laboratory or clinical-epidemiological criteria) occurring in Rio de Janeiro municipality between January and December 2016, by week and neighbourhood of the patient's residence.

Case definitions follow the Ministry of Health protocols. A suspected case of chikungunya is defined as a patient with sudden fever of over 38.5˚C and severe arthralgia or arthritis not explained by other conditions, and who either lives in endemic areas or has visited one up to

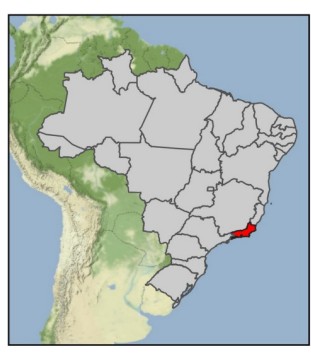

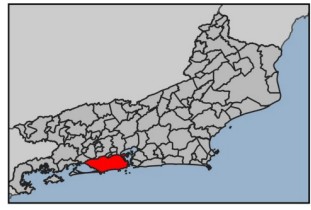

Sources: Instituto Pereira Passos and IBGE, Brazil; Stamen Design/ OpenStreetMap.

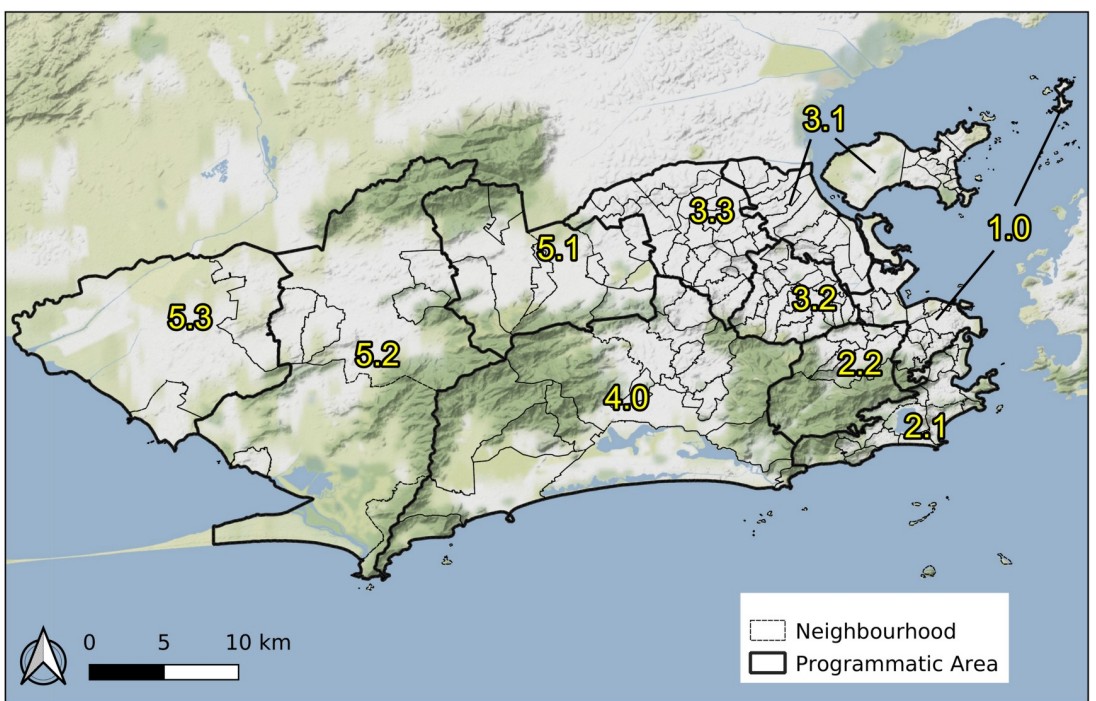

**Fig 2. Rio de Janeiro city by programmatic areas and neighbourhoods, 2010, Brazil.** Map created using QGIS version 3.12. Map layers by Instituto Pereira Passos (https://www.data.rio/) and Instituto Brasileiro de Geografia e Estatística (https://www.ibge.gov.br/). Map tiles by Stamen Design (http://maps.stamen.com/), under CC BY 3.0. Data by OpenStreetMap.

two weeks before the onset of symptoms or has an epidemiological link with a confirmed case. A confirmed case is a suspected case with at least one positive specific laboratory test for CHIKV or confirmed by clinical-epidemiological criteria [30].

**Socioeconomic data.**   To characterise the socioeconomic status, we obtained the sociode-velopment index data by neighbourhood from the *Instituto Pereira Passos* [31]. This index is based on eight indicators from the 2010 Demographic Census: 1) the percentage of domiciles with adequate water supply; 2) the percentage of domiciles with adequate sewage; 3) the per-centage of domiciles with garbage collection; 4) the average number of toilets per resident; 5) the percentage of illiteracy among residents between 10 and 14 years old; 6) per capita income of the domiciles, expressed as minimum wages; 7) the percentage of domiciles with per capita income up to one minimum wage, and 8) the percentage of domiciles with per capita income greater than five minimum wage. The sociodevelopment index is calculated as the arithmetic average of the normalised indicators (each ranging from 0 to 1, being 0 the worst socioeco-nomic condition and 1, the best) [31].

We also obtained data on the population by neighbourhood from the *Instituto Pereira Passos* [32].

**Environment and temperature data.**   Land use data for the city of Rio de Janeiro were obtained from the *Instituto Pereira Passos* as a shapefile [33]. We created the category "green areas" by aggregating: agricultural areas, areas with swamps and shoals, areas with tree and shrub cover, and areas with woody-grass cover. After that, we calculated the proportion of green areas for each neighbourhood.

Temperature information for 2016 was obtained from 38 meteorological weather stations in Rio de Janeiro, from five different meteorological and environmental institutes. The institutes are the Brazilian National Institute of Meteorology [34], the Brazilian Airspace Control Department [35],

the Rio de Janeiro State Environmental Institute [36], the Rio de Janeiro Municipal Environmental Secretariat [37] and the *Alerta Rio* System [38]. All measurements are made according to the recommendations of the World Meteorological Organization [39]. The institutes make their meteorological data publicly available, being that the frequency of measurements of the first four organisations is hourly, while the *Alerta Rio* System measurements are made every 15 minutes.

From the temperature measurements, we computed the daily maximum, minimum and mean temperature. We also evaluated the availability of daily data in terms of missing measurements. The daily records that had more than 60% of missing measurements were excluded. We decided to use the minimum temperature as in tropical climates it acts as a limiting factor for the *Ae. aegypti* activity and population [40,41]. We then obtained the minimum temperature for each week and station, and, to obtain the minimum temperature by neighbourhood, we applied universal kriging. Briefly, kriging is a method that uses a sample of data points to estimate the value of a given variable over a continuous space [42]. First, we interpolated the minimum temperature to a grid with each unit measuring 500m x 500m. The grid with the meteorological weather stations locations is displayed in the S1 Fig. Then we obtained the minimum temperature of the neighbourhood by calculating the average of the minimum temperature of the grid units whose centroids were within the boundaries of the neighbourhood.

To process and organise the environmental data, we used R version 3.6.1 [43] and packages sf [44], geoR [45] and tidyverse [46].

## Statistical analysis

We used the Stan platform [47,48] to fit spatio-temporal models, more specifically ICAR models, to a dataset consisting of neighbourhoods counts of chikungunya cases, exploring the relationship with sociodevelopment index, the proportion of green areas and minimum temperature. Let $Y_{i,t}$ be the counts of chikungunya cases at neighbourhood $i = 1, 2, \ldots,$ n = 160, and week $t = 1, 2, \ldots,$ T, where $Y_{i,t} \sim Poisson(\mu_{i,t})$. We explored different structures for $\mu_{i,t}$, presented in the S1 Appendix along with each Watanabe-Akaike information criterion (WAIC) [49]. The model selected based on the WAIC has the following structure:

$$log(\mu_{i,t}) = log(e_i) + \beta_0 + X'_i\beta_{k,t} + U_{i,t} + \phi_i$$
$$U_{i,t} = \rho_i U_{i,t-1} + \zeta_i Temperature_{i,t}$$

(1)

The latent spatial effect is represented by $\phi$, which, a *priori*, follows a conditional autoregressive distribution [24]; that is, the conditional distribution of each $\phi_i$ follows a normal distribution whose mean and variance depend on the neighbourhood structure $w_{ij}$. Assuming a binary neighbourhood structure, where $w_{ij} = 1$ if areas $i$ and $j$ share borders and 0 otherwise, each $\phi_i$ follows a conditional normal distribution whose mean is equal to the average of its neighbours, and its variance is inversely proportional to the number of neighbours $d_i$, that is:

$$p\left(\phi_i | \phi_{i\sim j}\right) = N\left(\frac{\sum_{i\sim j}\phi_i}{d_i}, \frac{\sigma^2}{d_i}\right)$$

(2)

The expected number of chikungunya cases at neighbourhood $i$ ($e_i$) represents the number of cases that would have been observed if there were no differences in the incidence of cases across time and space:

$$e_i = \left(\frac{\sum_{i=1}^{n}\sum_{t=1}^{T}Y_{i,t}}{\sum_{i=1}^{n}population_i}population_i\right)/T,$$

(3)

$\beta_0$ is the intercept, $X'_i$ represents a vector of $k$ covariates and $\beta_{k,t}$ is the coefficient of covariate $k$ in week $t$. We decided to allow for time-varying coefficients for the covariates to explore if their association with the number of cases vary as the epidemic progresses. The covariates included in the $X'_i$ vector were the sociodevelopment index and the proportion of green areas. The proportion of green areas showed a skewed distribution. Therefore, this variable was transformed to the cubic root. We also fitted models including the population density. However, the 90% credible interval (CI) of the population density coefficient included 0 for most weeks, and the inclusion of this variable increased the WAIC. Therefore, the population density was not considered in the final model.

We estimated the temperature effect in a non-linear fashion using a transfer function ($U_{i,t}$), considering that the temperature has an immediate effect ($\zeta_i$) and that a proportion ($\rho_i$) of this effect propagates to future times. This proportion $\rho_i$ is called the memory effect and we considered it to be any value between 0 and 1 [26]. To combine and visualise both effects of the temperature, we obtained the impulse response function of the temperature for each neighbourhood. This function expresses the effect of a 1 unit increase in the temperature of one week propagating in time [26]. The temperature was standardised.

The models were fitted under the Bayesian framework using the Stan platform [47,48] to run four chains of 10000 iterations each where the first 5000 were the warmup. Relatively vague, proper prior distributions were used. We visually inspected the chains and used the R-hat statistic to check convergence [47,50,51]. Model validation was performed by comparing the fitted number of cases (mean and 90% CI) with the observed number of cases [49]. We also computed the Bayesian p-values of the different models and the nonrandomized Probability Integral Transform (PIT) [52] to investigate model adequacy (S2 Appendix). It is worth mentioning that we also fitted models that considered the reparametrisation of the Besag-York-Mollié (BYM2) as proposed by Riebler et al. [53]. However, the random component was over 90% spatial, and the unstructured effect 90% CI included zero for all neighbourhoods.

For the statistical analysis, we used R version 3.6.1 [43] and packages RStan [54] and loo, which was used to obtain the WAIC [55]. The R script and models codes in Stan are available at https://github.com/laispfreitas/ICAR_chikungunya [56]. Maps and graphs were created using QGIS version 3.12 [57] and ggplot2 version 3.2.0 [58].

## Results

Between January and December 2016, 13,627 cases of chikungunya were notified in the city of Rio de Janeiro, corresponding to an incidence of 21.6 cases per 10,000 inhabitants. The number of cases peaked at week 17/2016, with 1118 chikungunya cases (Fig 3A). The cumulative number of cases by neighbourhood ranged from 0 (Grumari, area 4.0) to 721 (Realengo, area 5.1). The highest incidence was found in Catumbi (area 1.0), of 211.0 cases per 10,000 inhabitants (Fig 3B).

The mean sociodevelopment index was 0.6080, ranging from 0.282 in Grumari (area 4.0) to 0.819 in Lagoa (area 2.1). Higher sociodevelopment indexes were observed in areas 2.1 and 4.0 (Fig 4A). Fifteen neighbourhoods did not have any green areas, mostly located in areas 1.0, 3.1, 3.2 and 3.3 (Fig 4B). Alto da Boa Vista (area 2.2) presented the highest percentage of green areas, of 90.4%. The average minimum temperature was 19.9˚C, ranging from 10.7˚C in Campo dos Afonsos (area 5.1) to 26.1˚C in Cidade Nova (area 1.0). Neighbourhoods in the east coastal region of Rio had higher temperatures on average (Fig 4C). The temperature decreased in the city around week 17, starting to increase again around week 35 (Fig 4D).

**A**

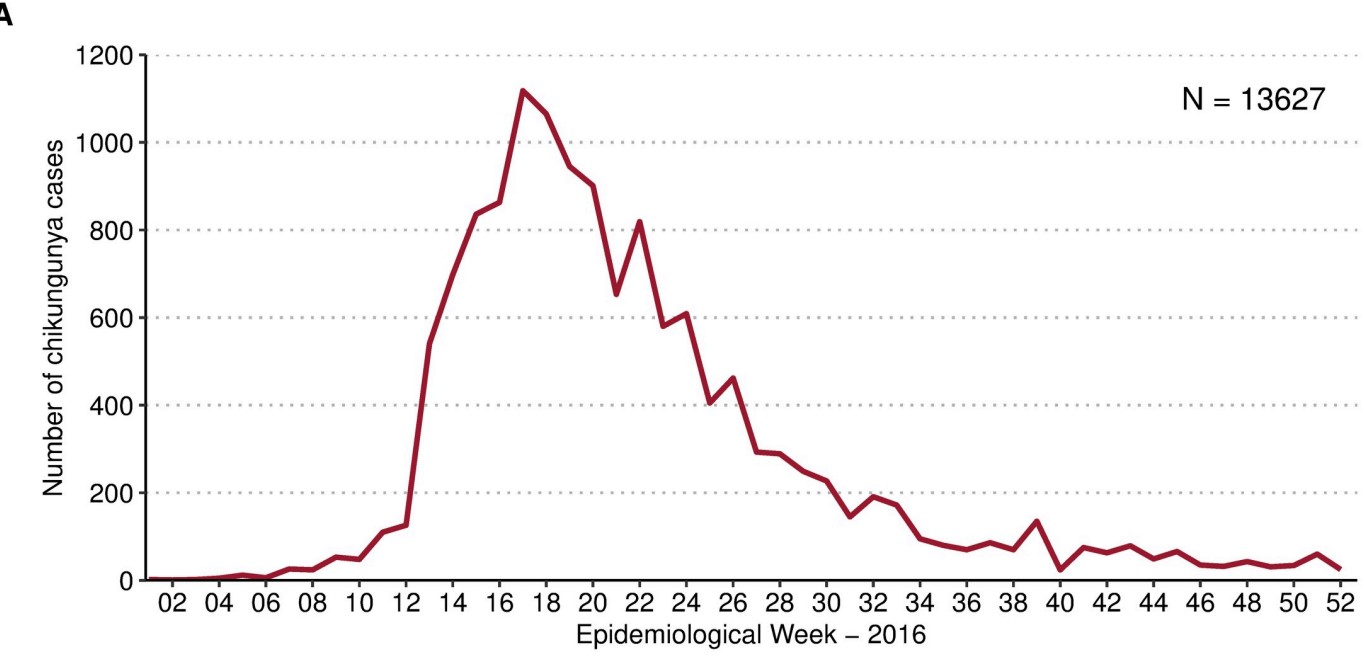

**B**

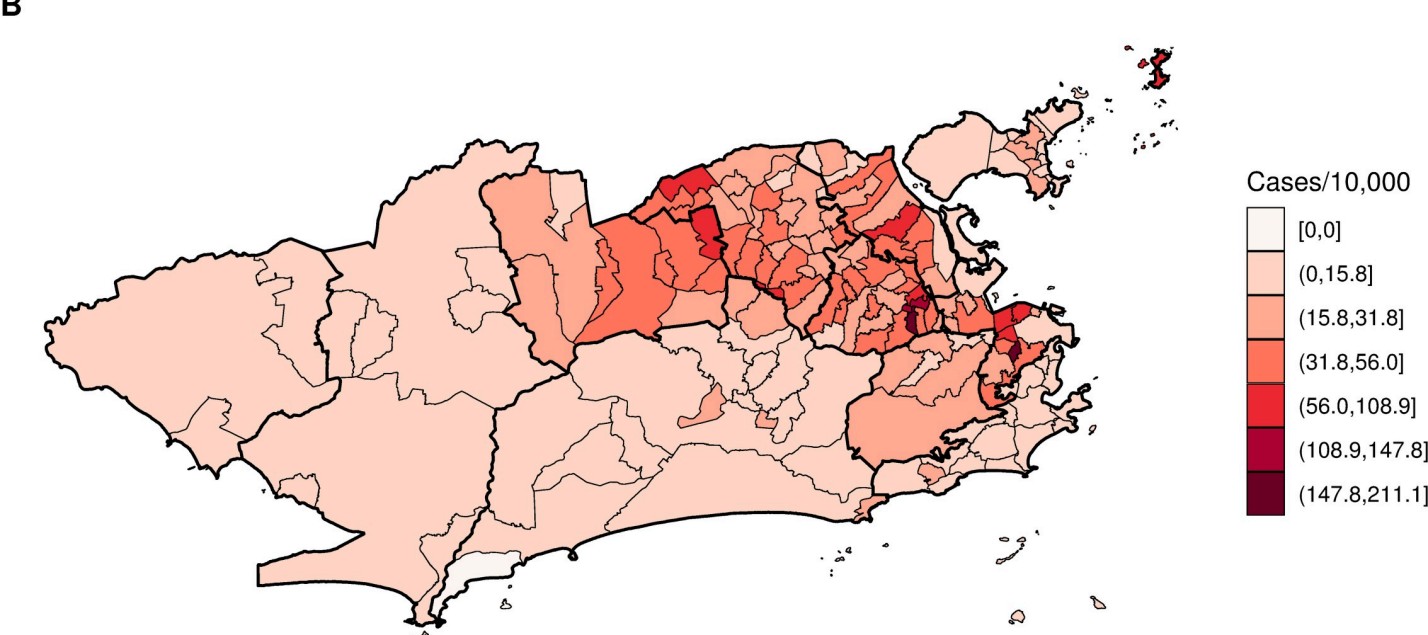

**Fig 3. Notified chikungunya cases by week (A) and chikungunya cases cumulative incidence per 10,000 inhabitants by neighbourhood (B), January to December 2016, Rio de Janeiro city, Brazil.** Map created using R version 3.6.1. Map layers by Instituto Pereira Passos (https://www.data.rio/).

Due to the small numbers of chikungunya cases at the beginning of 2016, we decided to model the cases starting at week 9, when the number of cases in the city exceeded 50 for the first time.

The posterior summary (mean and 90% CI) of the time-varying coefficients for the sociodevelopment index and the proportion of green areas is presented in Fig 5. The

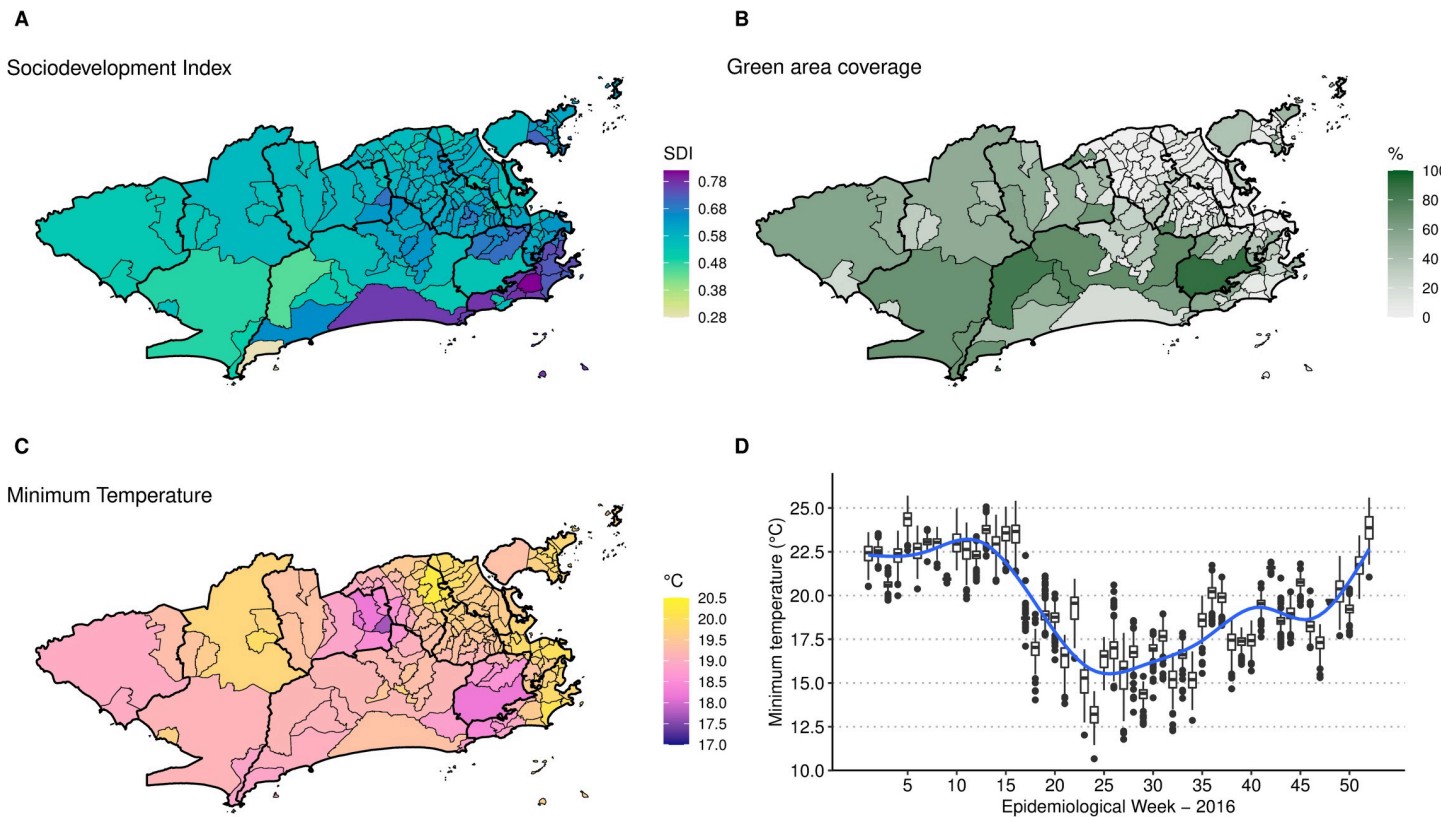

**Fig 4. Sociodevelopment index in 2010 (A), percentage of green areas in 2015 (B), minimum temperature (˚C) average in 2016 by neighbourhood (C) and boxplot of the minimum temperature (˚C) by neighbourhood and week (D), Rio de Janeiro city, Brazil.** Maps created using R version 3.6.1. Map layers by Instituto Pereira Passos (https://www.data.rio/).

sociodevelopment index presented a strong protective effect at the beginning of the epidemic. The sociodevelopment index effect was inversely associated with the epidemic curve: as the number of cases increased, the protective effect decreased, remaining almost constant during the peak of the epidemic, and increasing again once the number of cases started decreasing. The effect of the sociodevelopment index was null during the peak of the epidemic (around week 17). The proportion of green areas effect included 0 in the 90% CI in most of the weeks. However, when the spatial component was not included in the model, it presented a protective effect (S2C Fig).

The latent spatial structure has a clear trend of positive spatial effects in areas where the epidemic was concentrated (areas 1.0, 2.2, the mainland part of 3.1, 3.2, 3.3 and 5.1) and negative spatial effects in less affected areas (Fig 6). The inclusion of the covariates in the final model decreased the spatial effects in 102 of the 160 neighbourhoods compared to the model with only the spatial effect (S3 Fig).

The posterior distributions (mean and 90% CI) of the instantaneous and memory effects of the minimum temperature are displayed in Fig 7. For most neighbourhoods (113/160, or 70.6%) the instantaneous effect of the temperature increased the risk of chikungunya cases (Fig 7A). The instantaneous temperature effect, however, was in general small, reaching its maximum in Catumbi (area 1.0), where the temperature relative risk was 2.28 (90%CI 2.07–2.53). The memory effect (Fig 7B) represents the propagation in time of the instantaneous effect. Therefore, in neighbourhoods where the instantaneous temperature effect was null, the memory effect is irrelevant.

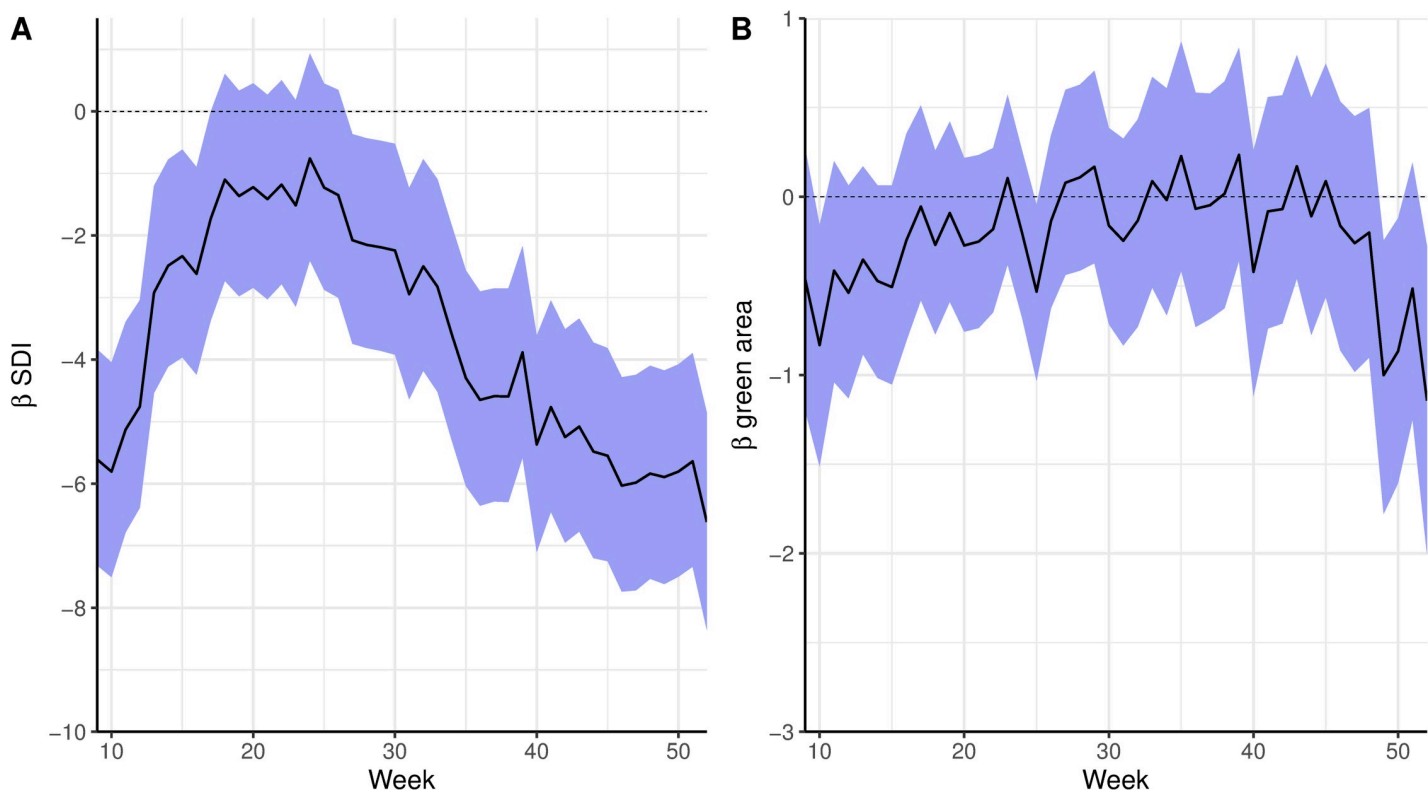

**Fig 5.** Time-varying coefficients (in the log scale, mean and 90% credible interval) for sociodevelopment index (SDI) (A) and proportion of green areas (B) for a spatial model for chikungunya cases from weeks 9 to 52 2016 and controlling for minimum temperature, Rio de Janeiro city, Brazil.

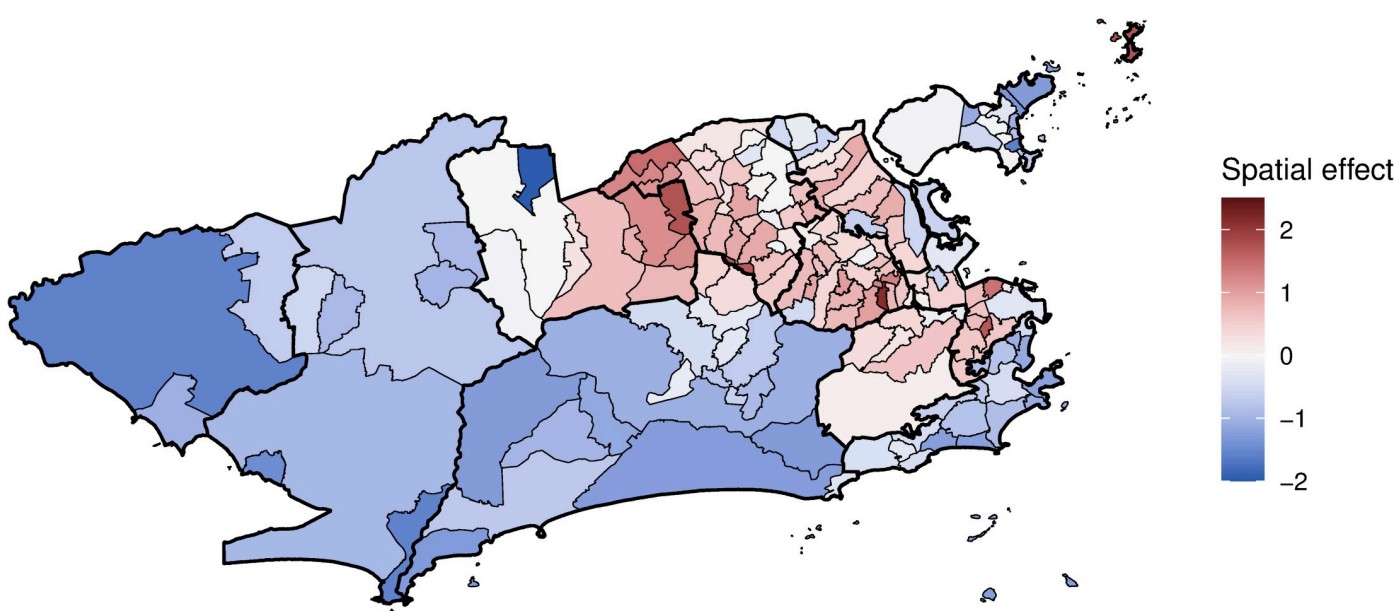

**Fig 6. Chikungunya cases mean spatial effects (in the log scale) controlling for sociodevelopment index, proportion of green areas, and minimum temperature, weeks 9 to 52 2016, Rio de Janeiro city, Brazil.** Map created using R version 3.6.1. Map layers by Instituto Pereira Passos (https://www.data.rio/).

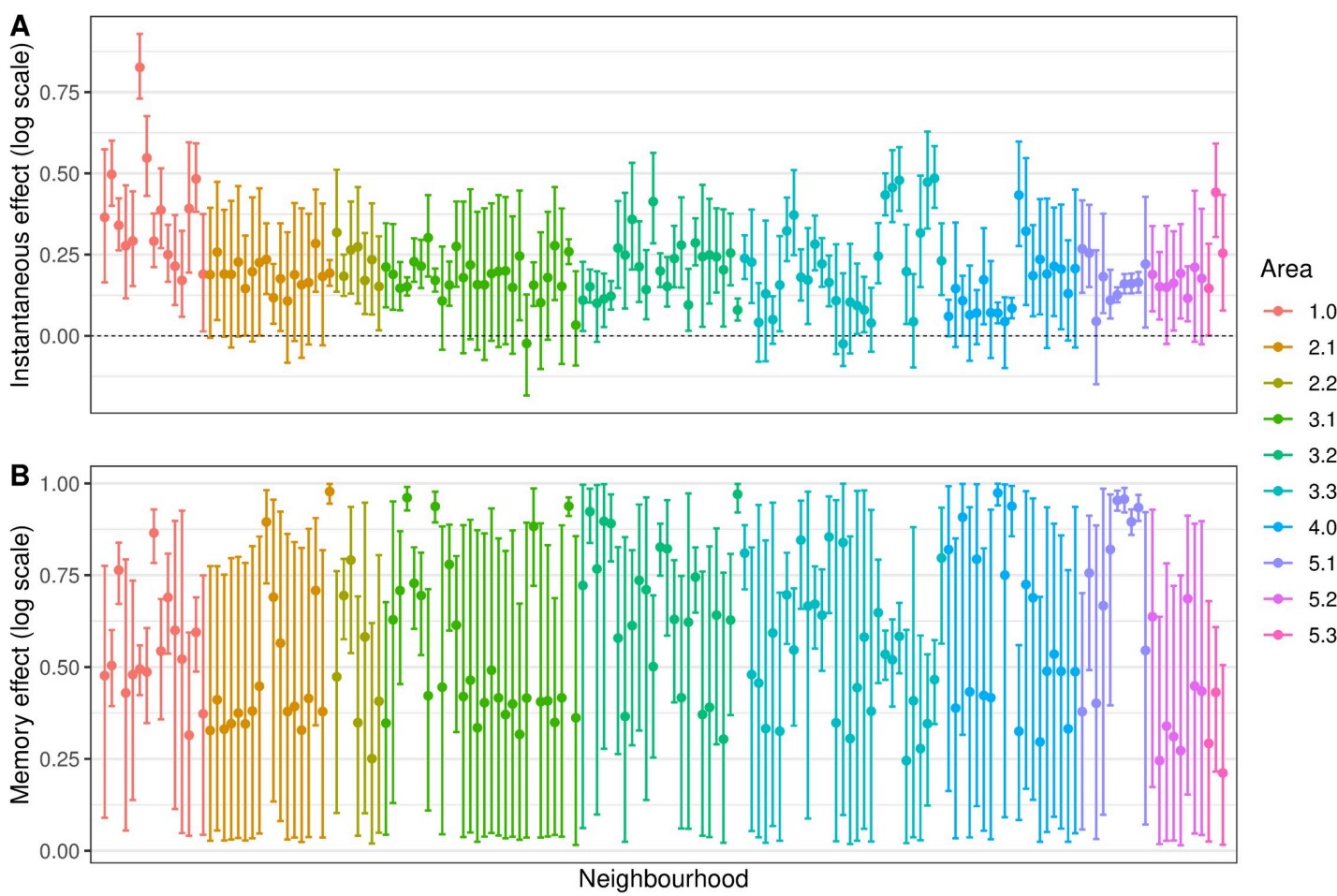

**Fig 7.** Minimum temperature instantaneous effect (in the same week) (A) and memory effect (B) on chikungunya cases (in the log scale) by neighbourhood, mean and 90% credible interval, controlling for sociodevelopment index and proportion of green areas, and the latent spatial effect, weeks 9 to 52 2016, Rio de Janeiro city, Brazil.

The combined effect of the minimum temperature, represented by the impulse response function, presented three patterns. These patterns are exemplified with nine selected neighbourhoods in Fig 8: null effect (Fig 8 first row), rapid decay of the effect (second row), and slow decay of the effect (third row). The impulse response functions for all neighbourhoods are available in the S4 Fig.

The impulse response function is represented in time and space in Fig 9 and S1 Video. The first map depicts the mean temperature instantaneous relative risk, the impulse. The following maps show the propagation of the impulse on subsequent weeks. When the temperature relative risk is null (90% CI includes the 1), the neighbourhood is depicted blank. The strong memory effect in some neighbourhoods (Fig 7B) is observed by the persistence of the temperature effect for several weeks after the impulse. However, such effect declines to values very close to 1. These neighbourhoods were concentrated in areas 1.0, 2.2, mainland 3.1, 3.2, 3.3 and 5.1.

The mean estimated chikungunya relative risk increased rapidly in the first weeks, peaking at week 17 and then decaying progressively (S2 Video and Fig 10). The decrease in the chikungunya relative risk coincided with the decrease in the minimum temperature in the city (Fig 4D). High relative risks for chikungunya were mostly observed in areas 1.0, 2.2, 3.1, 3.2, 3.3

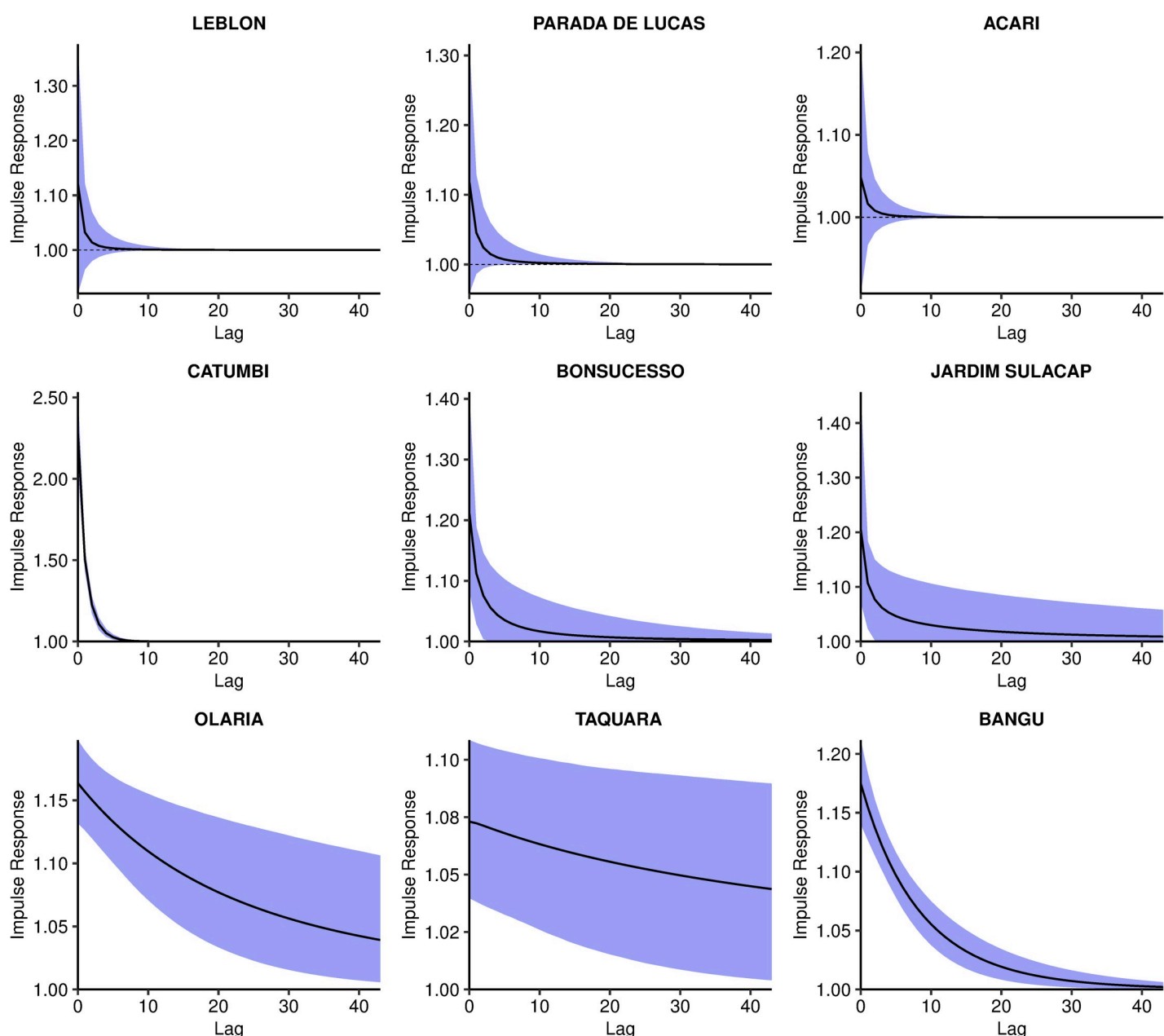

**Fig 8. Impulse response function of the minimum temperature effect on chikungunya cases over time, posterior mean and 90% credible interval, controlling for sociodevelopment index and proportion of green areas, and the latent spatial effect, in selected neighbourhoods, Rio de Janeiro city, Brazil.**

and 5.1. The neighbourhoods of the remaining areas presented chikungunya relative risks below 1 for almost the entire study period. The classification of the neighbourhoods in terms of chikungunya relative risk considering the 90% CI is available in the S5 Fig.

## Discussion

In this study, we estimated the associations of socioeconomic status, environment, and temperature with the spatio-temporal distribution of the first chikungunya epidemic in Rio de Janeiro city. The sociodevelopment index and the proportion of green areas were included in

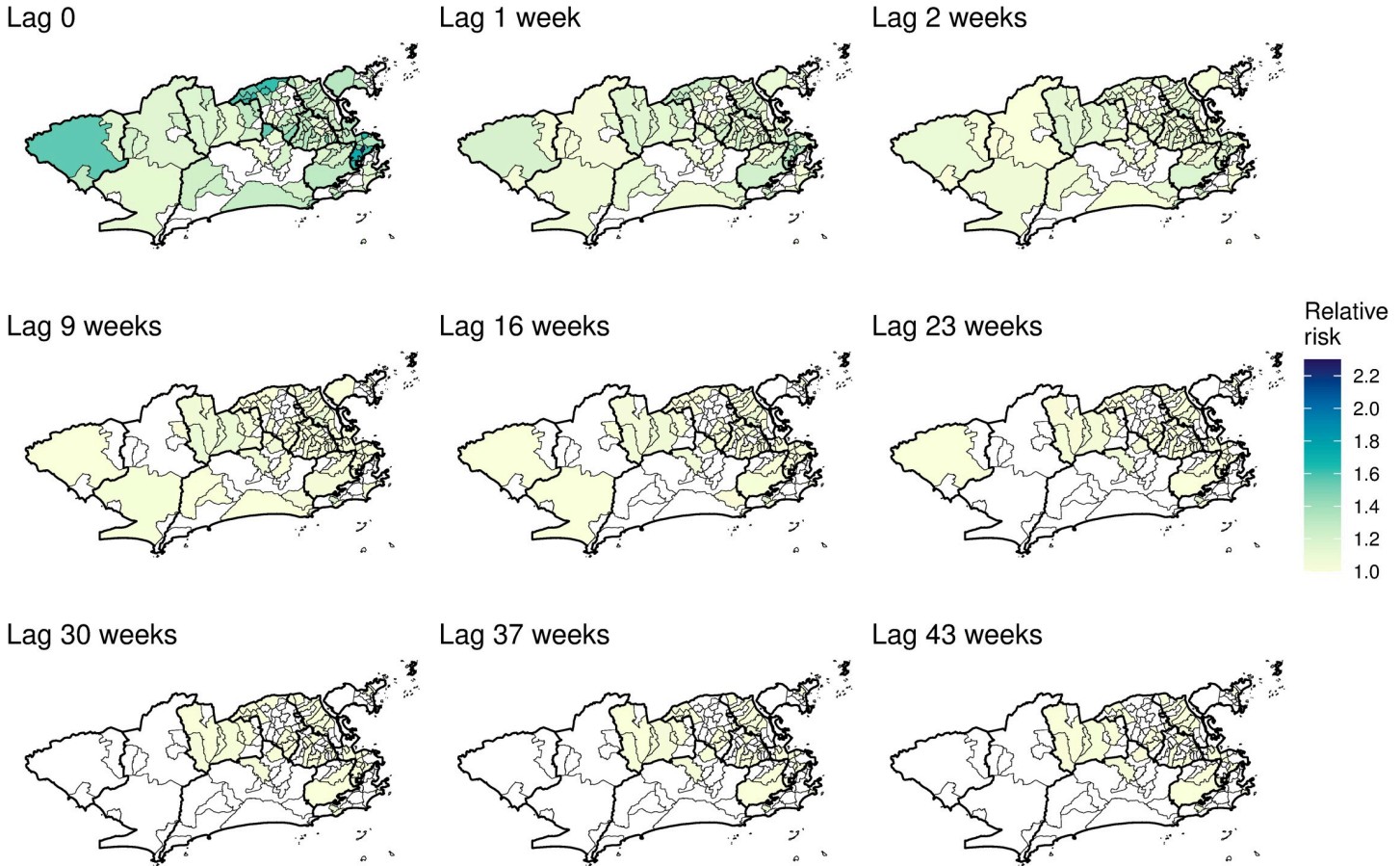

**Fig 9. Minimum temperature instantaneous effect on chikungunya cases and its propagation in time by neighbourhood in selected weeks, controlling for sociodevelopment index and green areas proportion, and the latent spatial effect, Rio de Janeiro city, Brazil.** Maps created using R version 3.6.1. Map layers by Instituto Pereira Passos (https://www.data.rio/).

the model with time-varying coefficients, allowing us to explore how the effects of these factors changed throughout the epidemic. The temperature was included in the model in a non-linear fashion using a transfer function, considering that the temperature has an immediate effect and that a proportion of this effect propagates to future times. To the best of our knowledge, this is the first time a transfer function is applied for temperature when modelling *Aedes*-borne diseases.

The sociodevelopment index was inversely associated with the risk of chikungunya in all models (Figs 5A and S2A and S2B). In fact, the sociodevelopment index presented the strongest effect in the models, which strengthens the hypothesis of chikungunya being a disease of social determination. This index is composed of sanitary conditions indicators, among others, and poor sanitary conditions are known to favour the reproduction of the *Ae. aegypti* mosquitoes. The association of low socioeconomic locations with increased risk of chikungunya was also found in a study in French Guiana [59] and a study in Barraquilla, a Colombian city [22]. Our results indicate that poor neighbourhoods were affected first and hardest by the chikungunya epidemic, highlighting the importance of vector control activities in socioeconomically vulnerable locations.

When the spatial dependency was not included in the model, the proportion of green areas was negatively associated with the number of chikungunya cases (S2C Fig). Such association

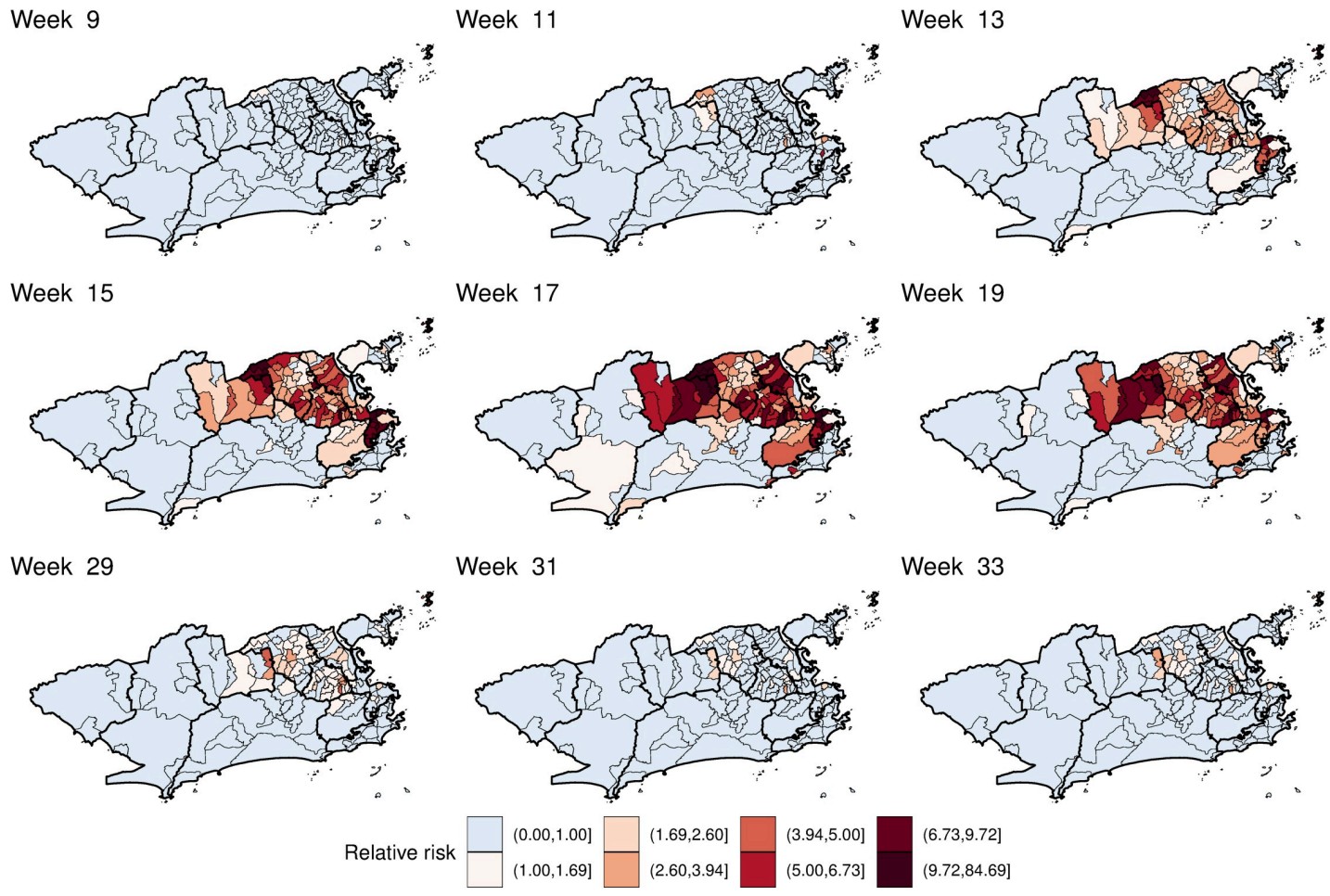

**Fig 10. Posterior mean chikungunya relative risk by neighbourhood in selected weeks, controlling for sociodevelopment index, proportion of green areas, and minimum temperature, and the latent spatial effect, Rio de Janeiro city, Brazil.** Maps created using R version 3.6.1. Map layers by Instituto Pereira Passos (https://www.data.rio/).

was observed for dengue in São Paulo, where low vegetation cover areas presented higher dengue incidence rates [23]. However, with the inclusion of the spatial component, the effect of the proportion of green areas moved towards the null. This is possible due to spatial confounding, which happens when covariates that are spatially smooth are collinear with spatial random effects [60]. In Rio de Janeiro, the majority of the green areas are in the mountain massifs (Fig 2), which trespasses different neighbouring borders.

The temperature was associated with an increase in the risk of chikungunya in most neighbourhoods. In our models, we assumed that the temperature presents an effect that is a combination of an instantaneous effect and of a proportion of this that propagates in time (the memory effect). Although the instantaneous association is relatively low across the different neighbourhoods, it can persist for a long period of time for some of the districts. This was only possible because we allowed the parameters present in the transfer function to change across neighbourhoods. Different areas experiencing heterogeneous temperature effects for *Aedes*-borne diseases were also observed in previous studies [61,62]. The inclusion of space-varying coefficients for the temperature improved the fitting of our models.

The instantaneous association represents the effect of the temperature on the activity of the mosquito, human behaviour and the mosquito-virus interaction. The biting rate of the *Ae.*

*aegypti* increases with the temperature until around 35˚C [10], and people become more exposed to mosquitoes in warm temperatures. The temperature also accelerates the virus extrinsic incubation period in the mosquito, and transmission was estimated to peak around 28.5˚C [10]. On the other hand, the temperature affects the population of mosquitoes by increasing fecundity, egg-to-adult survival, development rate and lifespan [10]. This effect is not only on the same week but also accumulates in time, which is captured by the memory effect.

Interestingly, the CHIKV epidemic in Rio de Janeiro in 2016 did not reach the whole city, with high-risk areas mostly concentrated in the North and Downtown regions. The decrease in the number of cases coincided with the drop in the minimum temperature, around week 17 (Figs 3A and 4D). These two observations combined suggest that the epidemic was interrupted not because of susceptible human population depletion in the city, but because the drop in temperature caused a reduction in the transmission in such a way that the epidemic was not sustained. It is important to note that although the number of cases diminished substantially, there were still chikungunya cases being reported until the end of the year. Rio de Janeiro is a tropical city, and the minimum temperature rarely is below the minimum temperature needed for transmission to occur, of 13.5˚C [10]. A previous study conducted in the city showed that the *Ae. aegypti* population varies seasonally, but the mosquito is endemic all over the year [63]. This could explain the long persistence of the temperature effect in time in some neighbourhoods (Fig 9 and S1 Video).

The application of spatial models considering intra-urban scenarios is still growing for *Aedes*-borne diseases. Our study identified high-risk neighbourhoods for the first chikungunya epidemic in Rio de Janeiro city, which concentrated mostly in the North and Downtown regions (Fig 10 and S2 Video). Such regions were already identified as high-risk locations for dengue [64]. In our previous study, neighbourhoods from these regions were more likely to constitute simultaneous clusters for dengue, Zika and chikungunya [65]. These regions have a combination of factors that favours the *Ae. aegypti*'s ecology and transmission of diseases: low vegetation, low socioeconomic status and increased temperature (Fig 4). Vector control activities should be prioritised and intensified in the identified high-risk areas, as they also appear to be the first ones affected by the epidemic. Additionally, the long persistence of the temperature effect (Fig 9 and S1 Video) and a small number of cases even after the decline of the epidemic, indicate that the mosquito continues to circulate and transmit the disease throughout the year. Therefore, interventions must be designed to be continuous and to work in the long term in these locations.

Our study has some limitations. As for any study using passive surveillance data on *Aedes*-borne diseases, there is an uncertainty on the diagnosis of the reported cases as well as under-reporting. Because the data are generated from suspected cases attending health care facilities, asymptomatic and mild cases who do not seek medical assistance are usually not detected. However, SINAN has been implemented for decades, representing an important and robust data source for the study of *Aedes*-borne diseases in Brazil [66]. It should be noted that, in the same year, the city of Rio de Janeiro was also experiencing dengue and Zika epidemics [65]. Because of the association between Zika and severe congenital manifestations, the disease awareness around Zika may have improved the search for medical care and the reporting rates [67]. On the other hand, the simultaneous occurrence of three arbovirus epidemics may have impaired the differential diagnosis, as they cause similar symptoms. Another limitation is the spatial unit. We analysed the data aggregated at the neighbourhood level. Although the data are more reliable at this spatial unit, smaller areas inside the same neighbourhood can present different socioeconomic and environmental characteristics. Finer scales such as census tracts should be considered in future studies. Finally, an important limitation is the assumption that

the chikungunya risk is related to the neighbourhood of residence, while some people may get infected in other locations. This is an unavoidable limitation when using surveillance data. It is not a trivial task to identify where the person was infected when dealing with *Aedes*-borne diseases. Entomological surveillance research and human population mobility data could be explored in future studies and potentially bring insight on where the most common places of transmission are.

The model here presented has the potential to be applied to other cities and other urban *Aedes*-borne diseases. Mosquito population information is expensive to collect and often unreliable. Therefore not depending on such data is a strength of our model. Another strength is the application of a transfer function to estimate the non-linear effect of the temperature and its duration. By using temperature, socioeconomic status and proportion of green areas data as proxies of the key elements of CHIKV transmission, our model contributed to the better understanding of the spatio-temporal dynamics associated with first chikungunya epidemic in a tropical metropolitan city. Importantly, our results indicate that even considering the environment and the temperature, the socioeconomic status plays a major role in affecting the incidence and distribution of a first *Aedes*-borne disease epidemic in a large city. This strengthens the importance of improving sanitary conditions and taking measures to diminish social inequality as necessary actions to control and prevent *Aedes*-borne diseases.

## Supporting information

**S1 Fig. Meteorological weather stations (red dots) in the 500m X 500m grid and neighbourhoods, Rio de Janeiro city, Brazil.** Maps created using R version 3.6.1. Map layers by Instituto Pereira Passos (https://www.data.rio/).
(PNG)

**S2 Fig.** Time-varying coefficients (in the log scale, mean and 90% credible interval) for socio-development index (SDI) (A,B) and proportion of green areas (C,D) without (model 1) and with (model 2) spatial dependency, for chikungunya cases from weeks 9 to 52 2016, Rio de Janeiro city, Brazil.
(PNG)

**S3 Fig. Correlation between the spatial effects (in the log scale) of Model 0 versus Model 4, by neighbourhood, weeks 9 to 52 2016, Rio de Janeiro city, Brazil.**
(PNG)

**S4 Fig. Impulse response of the minimum temperature, posterior mean and 90% credible interval, by neighbourhood, Rio de Janeiro city, Brazil.**
(PDF)

**S5 Fig. Classification of the chikungunya relative risk by neighbourhood in selected weeks based on the 90% credible interval (controlling for sociodevelopment index, proportion of green areas and minimum temperature, and the latent spatial effect), Rio de Janeiro city, Brazil.** Risk: 90%CI >1. Protection: 90%CI <1. None: 90%CI includes 1. Maps created using R version 3.6.1. Map layers by Instituto Pereira Passos (https://www.data.rio/).
(PNG)

**S1 Video. Minimum temperature instantaneous effect on chikungunya cases and its propagation in time by neighbourhood, controlling for sociodevelopment index and proportion of green areas, and the latent spatial effect, Rio de Janeiro city, Brazil.** Maps created using R version 3.6.1. Map layers by Instituto Pereira Passos (https://www.data.rio/).
(MP4)

**S2 Video. Posterior chikungunya relative risk by neighbourhood, controlling for sociode-velopment index, proportion of green areas and minimum temperature, and the latent spatial effect, weeks 9 to 52 2016, Rio de Janeiro city, Brazil.** Maps created using R version 3.6.1. Map layers by Instituto Pereira Passos (https://www.data.rio/).
(MP4)

**S1 Appendix. Models structures explored to estimate μi,t and each Watanabe-Akaike information criterion (WAIC).**
(PDF)

**S2 Appendix. Model checking.**
(PDF)

## Acknowledgments

The authors would like to thank the Municipal Secretariat of Health for providing the data on reported cases, and the meteorological and environmental institutes (INMET, DECEA, INEA, SMAC and *Alerta Rio*), for making their meteorological data publicly available.

## Author Contributions

**Conceptualization:** Laís Picinini Freitas, Alexandra M. Schmidt, Oswaldo Gonçalves Cruz, Marilia Sá Carvalho.

**Data curation:** Laís Picinini Freitas, William Cossich, Oswaldo Gonçalves Cruz.

**Formal analysis:** Laís Picinini Freitas, Alexandra M. Schmidt, Oswaldo Gonçalves Cruz, Marilia Sá Carvalho.

**Investigation:** Laís Picinini Freitas, William Cossich, Oswaldo Gonçalves Cruz.

**Methodology:** Laís Picinini Freitas, Alexandra M. Schmidt, Oswaldo Gonçalves Cruz, Marilia Sá Carvalho.

**Project administration:** Laís Picinini Freitas, Marilia Sá Carvalho.

**Software:** Alexandra M. Schmidt, Oswaldo Gonçalves Cruz.

**Supervision:** Alexandra M. Schmidt, Oswaldo Gonçalves Cruz, Marilia Sá Carvalho.

**Validation:** Laís Picinini Freitas, Alexandra M. Schmidt.

**Visualization:** Laís Picinini Freitas, Oswaldo Gonçalves Cruz.

**Writing – original draft:** Laís Picinini Freitas.

**Writing – review & editing:** Alexandra M. Schmidt, William Cossich, Oswaldo Gonçalves Cruz, Marilia Sá Carvalho.

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
