## [Decision Letter · Decision Letter 0]

3 Apr 2021

Dear Dr Freitas,

Thank you very much for submitting your manuscript "Spatio-temporal modelling of the first Chikungunya epidemic in an intra-urban setting: the role of socioeconomic status, environment and temperature" for consideration at PLOS Neglected Tropical Diseases. As with all papers reviewed by the journal, your manuscript was reviewed by members of the editorial board and by several independent reviewers. The reviewers appreciated the attention to an important topic. Based on the reviews, we are likely to accept this manuscript for publication, providing that you modify the manuscript according to the review recommendations. 

Sincerely,

Pedro F. C. Vasconcelos

Deputy Editor

Pedro Vasconcelos

Deputy Editor

Reviewer's Responses to Questions

**Key Review Criteria Required for Acceptance?**

**Methods**

-Are the objectives of the study clearly articulated with a clear testable hypothesis stated?

-Is the study design appropriate to address the stated objectives?

-Is the population clearly described and appropriate for the hypothesis being tested?

-Is the sample size sufficient to ensure adequate power to address the hypothesis being tested?

-Were correct statistical analysis used to support conclusions?

-Are there concerns about ethical or regulatory requirements being met?

Reviewer #1: The data source is introduced, but little discussion is given to the data generating mechanism. That is, how does the epidemic in the population relate to the observations. Is there evidence for asymptomatic and undetected transmission? Are there disparities in recording cases? This should be addressed. 

Line 221: It may not make a difference, but a quantitative convergence diagnostic like the Gelman and Rubin diagnostic is preferable to visual assessment. 

Statistical Analysis Generally: I'd be interested to see a posterior predictive p-value to assess the final model fit. I think the model is meaningful and useful even if the fit is questionable, but it would be good to higlight this kind of limitation. 

The proposed model is not dynamic in the same way that compartmental techniques are, so this is one area where limitations of the proposed method may arise. 

I applaud the authors for making their code available.

Reviewer #2: (No Response)

**Results**

-Does the analysis presented match the analysis plan?

-Are the results clearly and completely presented?

-Are the figures (Tables, Images) of sufficient quality for clarity?

Reviewer #1: I found the results to be well articulated.

Reviewer #2: (No Response)

**Conclusions**

-Are the conclusions supported by the data presented?

-Are the limitations of analysis clearly described?

-Do the authors discuss how these data can be helpful to advance our understanding of the topic under study?

-Is public health relevance addressed?

Reviewer #1: Overall the conclusions are well stated and the limitations well articulated. One area where this should be expanded is the same as mentioned previously - the difference between the techniques proposed and mechanistic models which account for transmission dynamics. To be clear, I think the authors' approach is reasonable, but it should be placed in the context of the wider stochastic epidemic modelling literature.

Reviewer #2: (No Response)

**Editorial and Data Presentation Modifications?**

Reviewer #1: Overall, the manuscript is well written and clear, both in terms of methodology and problem description. There are a few minor language issues to double check, mainly consisting of issues of word choice and numerical agreement. Here are a few examples - overall the writing is quite good. 

Abstract

"Green" areas should be clarified. Environments with vegetation are likely heterogeneous with respect to the factors identified. 

Summary

- Line 42: "transmitted viruses"

- Line 45: "status plays"

- Line 47: "improving ... is"

Introduction

- Line 61: facilitating the establishment of a

- Line 63: citation needed - antibody tests can certainly be cross-reactive between these viruses

- Line 66: "conditioned by" -> "favorable to"

- Line 73: "data regarding the"

- Line 93: "favor contact" (remove "the"), "the human" -> "humans"

- Line 96: "disease epidemics"

- Line 98: "including intrinsic conditional"

Reviewer #2: (No Response)

**Summary and General Comments**

Reviewer #1: Overall this is a strong manuscript in need of a few additional areas of discussion, and ideally some investigation of model fit (beyond the relative measures of information criteria).

Reviewer #2: The authors present a study of the first chikungunya outbreak in Rio de Janeiro in 2016, and analyze this dataset of notified cases by neighborhood and week using Bayesian (intrinsic conditional autoregressive [ICAR]) spatio-temporal models to understand the drivers of this outbreak. They investigate effects of temperature, and include this as both a direct and a decaying effect, socioeconomic status, and green space. Covariate effects are allowed to vary over time, so that they can tease apart whether the importance of different factors changes as the epidemic progresses. I think this is a strong and helpful piece of work. The insights on how socioeconomic status of neighborhoods is tied to transmission early on in the pandemic, and as it fades, but is essentially absent during the peak, is particularly interesting. I offer the following comments in the hope they can improve the paper further:

Socioeconomic status was based on an index, which was an average of eight normalized indicators. I am always somewhat uncomfortable with indicator variables, because the interpretation of an arithmetic average of normalized values becomes difficult to interpret – why not use a subset of the indicators directly as variables? Would it not be much more helpful to know whether it is water supply that leads to vulnerability, or sewage or garbage collection, and how important these different factors are relative to each other?

Green areas: can you explain in more detail why you combined these various “natural” areas into green space for your analysis? Areas like agricultural land are not associated typically with Ae. aegypti, while something like canopy cover could certainly lead to more favorable microclimates for this species. (in other words, the composite variable might end up not having an effect because it combines both positive and negative land use elements).

Although the minimum temperature can certainly be important, so can the maximum (e.g., temperatures that are overly hot can impair mosquitoes as well). Why only pick the minimum here?

I appreciate the mention of limitations in the discussion. For the third of those (people potentially getting infected in neighborhoods other than where they live), I’d like to see some more discussion, as with the other limitations: what do we know about this, how likely is it, how would you change your modelling approach to account for it if you think it likely is important?

PLOS authors have the option to publish the peer review history of their article (what does this mean?). If published, this will include your full peer review and any attached files.

Reviewer #1: No

Reviewer #2: No

Figure Files:

Data Requirements:

Reproducibility:

References

---

## [Decision Letter · Decision Letter 1]

3 Jun 2021

Dear Dr Freitas,

We are pleased to inform you that your manuscript 'Spatio-temporal modelling of the first Chikungunya epidemic in an intra-urban setting: the role of socioeconomic status, environment and temperature' has been provisionally accepted for publication in PLOS Neglected Tropical Diseases.

Best regards,

Pedro F. C. Vasconcelos

Deputy Editor

Pedro Vasconcelos

Deputy Editor

Reviewer's Responses to Questions

**Key Review Criteria Required for Acceptance?**

**Methods**

-Are the objectives of the study clearly articulated with a clear testable hypothesis stated?

-Is the study design appropriate to address the stated objectives?

-Is the population clearly described and appropriate for the hypothesis being tested?

-Is the sample size sufficient to ensure adequate power to address the hypothesis being tested?

-Were correct statistical analysis used to support conclusions?

-Are there concerns about ethical or regulatory requirements being met?

Reviewer #1: The authors did a good job of responding to both my comments and those of the other reviewer.

**Results**

-Does the analysis presented match the analysis plan?

-Are the results clearly and completely presented?

-Are the figures (Tables, Images) of sufficient quality for clarity?

Reviewer #1: The figures generally appear pixelated, and should be regenerated as high resolution raster or as vector image files.

**Conclusions**

-Are the conclusions supported by the data presented?

-Are the limitations of analysis clearly described?

-Do the authors discuss how these data can be helpful to advance our understanding of the topic under study?

-Is public health relevance addressed?

Reviewer #1: My comments have been addressed, via the expanded discussion of the authors' techniques relative to alternatives.

**Editorial and Data Presentation Modifications?**

Reviewer #1: I recommend to accept this manuscript.

**Summary and General Comments**

Reviewer #1: (No Response)

PLOS authors have the option to publish the peer review history of their article (what does this mean?). If published, this will include your full peer review and any attached files.

Reviewer #1: No

---

## [Editor Report · Acceptance letter]

14 Jun 2021

Dear Dr Freitas,

We are delighted to inform you that your manuscript, "Spatio-temporal modelling of the first Chikungunya epidemic in an intra-urban setting: the role of socioeconomic status, environment and temperature," has been formally accepted for publication in PLOS Neglected Tropical Diseases.

Best regards,

Shaden Kamhawi

co-Editor-in-Chief

Paul Brindley

co-Editor-in-Chief
